# Optical Fiber Based Mach-Zehnder Interferometer for APES Detection

**DOI:** 10.3390/s21175870

**Published:** 2021-08-31

**Authors:** Huitong Deng, Xiaoman Chen, Zhenlin Huang, Shiqi Kang, Weijia Zhang, Hongliang Li, Fangzhou Shu, Tingting Lang, Chunliu Zhao, Changyu Shen

**Affiliations:** Institute of Optoelectronic Technology, China Jiliang University, Hangzhou 310018, China; 1700201401@cjlu.edu.cn (H.D.); p1904085202@cjlu.edu.cn (X.C.); p20040854033@cjlu.edu.cn (Z.H.); 11801400310@cjlu.edu.cn (S.K.); 1700603202@cjlu.edu.cn (W.Z.); lihongliang@cjlu.edu.cn (H.L.); shufangzhou@cjlu.edu.cn (F.S.); langtingting@cjlu.edu.cn (T.L.); zhaochunliu@cjlu.edu.cn (C.Z.)

**Keywords:** optical fiber biosensor, Mach-Zehnder Interferometer, APES detection, condensation reaction

## Abstract

A 3-aminopropyl-triethoxysilane (APES) fiber-optic sensor based on a Mach–Zehnder interferometer (MZI) was demonstrated. The MZI was constructed with a core-offset fusion single mode fiber (SMF) structure with a length of 3.0 cm. As APES gradually attaches to the MZI, the external environment of the MZI changes, which in turn causes change in the MZI’s interference. That is the reason why we can obtain the relationships between the APES amount and resonance dip wavelength by measuring the transmission variations of the resonant dip wavelength of the MZI. The optimized amount of 1% APES for 3.0 cm MZI biosensors was 3 mL, whereas the optimized amount of 2% APES was 1.5 mL.

## 1. Introduction

3-aminopropyl-triethoxysilane (APES) is widely used in combination with crosslinkers, owing to its reactivity with different functional groups, such as aldehyde, carboxylic acid, and epoxy [1,2]. In fiber-optic nucleic acid biosensors, the silanization process is the chemical reaction between APES and a hydroxyl group, which will form a covalent linkage [3,4,5,6,7]. Additionally, the covalent linkage can fix amino groups on the surface of the fiber-optic Mach–Zehnder interferometer (MZI). Because of the considerable stability of the covalent linkage, optical fiber modified with APES has found widespread application [8]. Specifically, APES is a type of coupling reagent that can link two functional groups together, such as hydroxy and aldehyde [9].

While the condensation reaction between APES and hydroxyl group is occurring on the surface of the MZI, APES will also react with water [10]. To prevent APES from reacting with water, toluene is usually used as the solvent to prepare the APES solution. However, when APES is used in experiments, especially in fiber-optic biosensor experiments, there is no clear rule or experience to determine the amounts and concentrations of the APES solution. Unsuitable amounts and concentrations often lead to failure of the experiments. Usually, in APES-based fiber-optic sensing experiments, the APES concentrations mostly range from 1% to 2%, and the immersion time ranges from 45 min to 2 h [11,12,13,14,15,16,17]. Therefore, it is valuable to determine suitable APES amounts and concentrations for the fabrication of optical fibers coated with well-proportioned APES films.

In this paper, an optical fiber-based MZI for APES detection was proposed. The different amounts and concentrations of APES on the fiber surface can be monitored by the MZI’s interference patterns. The relationship between the amount of APES and the resonance dip wavelength was obtained. Optimized amounts of 1% APES and 2% APES for fiber-optic biosensors were also obtained.

## 2. Experimental Methods and Principle

### 2.1. Sensor Fabrication and Principle

A schematic diagram of the APES fiber-optic sensor based on MZI is shown in Figure 1. A fiber-coupled broadband source (BBS) with the wavelength range of 1500–1620 nm is used as the input light source. The transmission spectrum of the MZI is recorded by an optical spectrum analyzer (OSA: AQ6317B, YOKOGAWA, Japan) with a wavelength resolution of 0.02 nm. The BBS is coupled into the single-mode fiber (SMF), connecting to the OSA.

In the paper, a developed core-offset fusion SMF structure was used for the advantages of low cost and easy operation. Figure 2 shows an MZI based on the core-offset fusion SMF structure, where Figure 2a is a partially enlarged drawing of the MZI, and Figure 2b shows the core-offset connect joints on the fusion splicer screen. The fiber used in the experiment was Corning^®^ SMF-28e+^®^ BB. The core and cladding diameters of the SMF were 9 μm and 125 μm, respectively. The core-offset connect joints showing on the fusion splicer screen had no built-in rule scale. So, the cladding diameter has been marked as 125 um in Figure 2b.

The core-offset structure was formed by using a commercial electric-arc fusion splicer (Fujikura FSM-60s). Along the *x* axis direction of SMF1, SMF2 was shifted downward by 4 to 5 μm with a length of 3.0 cm. SMF2 and SMF3 also adopted core-offset fusion. SMF3 moved upward by 4 to 5 μm along the *x* axis direction of SMF2 [18].

For the core-offset structure, the light from SMF1 cladding is partly coupled to the SMF2 core to excite the core-mode light, and the rest of the light from SMF1 cladding enters the SMF2 cladding to excite the cladding-mode light. After propagating through SMF2, the component of core-mode light of SMF2 and component of cladding-mode light of SMF2 couple to the SMF3 core together, forming a type of MZI [19].

In the MZI, the phase difference Δϕ between the lights propagating in SMF2 cladding and core can be expressed as [20],
(1)$$\Delta \phi =2\pi (n_{eff}^{co}-n_{eff}^{cl})L/\lambda$$
where $n_{eff}^{co}$ and $n_{eff}^{cl}$ respectively represent the effective refractive indices of core-mode and cladding-mode, λ is the center wavelength of the input light, and L is the effective length of the MZI. The intensities I of the interference light wave of the MZI can be expressed as,
(2)$$I=I_{1}+I_{2}+2\sqrt{I_{1}I_{2}}\cos(\Delta \phi )$$
where $I_{1}$ and $I_{2}$ represent the intensities of the cladding-mode light and the core-mode light, respectively. Additionally, the fringe visibility K of the interference pattern is,
(3)$$K=\frac{2\sqrt{I_{1}I_{2}}}{I_{1}+I_{2}}$$

It can be seen from Equation (3) that the smaller the difference between $I_{1}$ and $I_{2}$, the larger K, and consequently the interference peak becomes stronger. In the output interferogram, the resonant dip wavelength corresponding to the interference peak can be expressed as,
(4)$$\lambda_{r}=2(n_{eff}^{co}-n_{eff}^{cl})L/2k+1$$
where $\lambda_{r}$ represents the resonant dip wavelength and k represents the wave number. When the external environment changes, the interference peak changes significantly.

### 2.2. APES Film Fabrication

Ninety-eight percent 3-aminopropyl-triethoxysilane (APES) (China Beijing Solibao Technology Co., Ltd.) was stored at room temperature (RT). Ninety-eight percent toluene and KOH used in this study were of analytical grade.

First, the MZI was rinsed 3 times with absolute ethanol and deionized water, respectively, to eliminate residue and dirt on the surface of the optical fiber, then dried naturally. Second, it was immersed in 0.1 M KOH for 1 h to carry out hydroxylation and form a layer of film with hydroxylated group as shown in Figure 3a. Next, after generating the hydroxylated group, the MZI was silanized with APES to form a layer of film with amino group; the reaction schematic diagram is shown in Figure 3b. In the experiments, 1% APES and 2% APES were adopted to drip onto the optical fiber surface, respectively. The amount of both APES concentrations ranged from 0 to 7.5 mL in 1.5 mL steps. By measuring the transmission intensities of the resonant dip wavelengths of the MZI, the relationships between the APES concentrations and resonance dip wavelength transmission intensities were obtained.

## 3. Results and Discussion

### 3.1. Validation of the Dripping Method’s Feasibility

In order to verify that adhesion between the optical fiber and APES could be achieved in a short time, we recorded and analyzed the transmission spectra of MZI immersed in 1% APES toluene solution for 1 h. Figure 4a shows the transmission spectra of the MZI corresponding to the MZI immersion time with 1% APES, and Figure 4b shows the relationship between the MZI immersion time with 1% APES and the resonant wavelength’s transmission intensities. When the MZI was immersed in 1% APES, the transmission intensities decreased instantly due to the condensation reaction. There was no drift for 1 h following, which illustrated that adhesion between the optical fiber and APES could be achieved in a short time. Removing the MZI from the APES 1 h later, the transmission intensities increased owing to the change in external environment from liquid to air.

### 3.2. Validation of the Necessity of Hydroxylation

A comparative experiment was performed to verify that APES cannot be directly combined with bare optical fiber, but can be combined with hydroxyl-modified optical fiber. After cleaning the bare optical fiber with absolute ethanol and deionized water, 1% APES was directly dripped onto the MZI without hydroxyl group. Cleaning the MZI with deionized water again, the transmission spectra underwent a change consistent with that of the bare MZI in air, as shown in the Figure 5.

This phenomenon showed that if the fiber is not modified with hydroxyl, APES cannot be directly combined with the bare optical fiber. The spectra changed after adding APES. While cleaning the MZI with deionized water, the transmission spectra underwent a change consistent with that of the bare MZI in air, which meant that APES adsorbed onto a fiber surface without hydroxyl could be removed.

### 3.3. Determination of APES Amount

The refractive indices of 1% APES toluene solution and fiber cladding are 1.496 and 1.462, respectively. Therefore, when dripping 1% APES onto the optical fiber surface, the effective refractive index of the cladding mode $n_{eff}^{cl}$ increased. According to Equations (1) and (2), the phase difference Δφ decreased, while the interference light intensity I increased. Concerning the resonant wavelength of the MZI, when $n_{eff}^{cl}$ increases, the resonant wavelength $\lambda_{r}$ should decrease. Additionally, the resonant wavelength of the MZI should shift to the short wavelength. However, because the function of APES is adhesion, stress may be generated on the MZI surface, affecting the resonant wavelength’s drift.

Figure 6a shows the transmission spectra of 1% APES amounts from 0 to 7.5 mL. The response time was very fast; the spectrum drifted immediately after drying the APES, and it stabilized within 10 min. Because there are many interference resonant wavelengths in the Mach–Zehnder interference spectrum, one resonant wavelength (near 1610 nm for instance) was selected for sensing. The transmission intensity of the resonant wavelength of 1612 nm was chosen as the sample wavelength. Figure 6b shows the relationship between the 1% APES amount and the resonant wavelength’s extinction ratio. Obviously, there was no linear relationship between the amounts of APES and the extinction ratio at the resonant wavelength of 1602 nm. When the applied amount of APES was about 1.5 mL, the hydroxylated group on the optical fiber was almost saturated. Therefore, continued increases in the amount of APES did not result in significant fluctuations in transmission intensity compared to that obtained with 1.5 mL. Due to the gradual saturation of the condensation reaction, the transmission intensities increased significantly and tended to stabilize after dripping 3.0 mL of 1% APES, which illustrated that the optimal amount of 1% APES was 3.0 mL.

The refractive indices of 2% APES toluene solution and fiber cladding are 1.493 and 1.462, respectively. When 1.5 mL 2% APES was dripped onto the MZI surface, the effective refractive index of cladding mode $n_{eff}^{cl}$ also increased. Thus, the same principle could illustrate that the transmission intensities increased.

The transmission spectra (near 1600 nm, for instance) of 2% APES amounts ranging from 0 to 7.5 mL are shown in Figure 7a. The transmission intensity of the resonant wavelength of 1612 nm was chosen to reflect the variations. Figure 7b shows the relationship between the amounts of 2% APES and resonant wavelength transmission intensities. Due to the gradual saturation of the condensation reaction, the transmission intensities increased significantly and tended to stabilize after dripping 1.5 mL of 2% APES, which illustrated that the optimal amount of 2% APES was 1.5 mL.

### 3.4. SEM of the APES-Modified Optical Fiber Surface

The 1% APES film on the optical fiber surface was examined using scanning electron microscopy (SEM) to verify that adhesion between the optical fiber and APES could be achieved with the dripping method. Figure 8a is the 500 times magnified side view of the APES film after soaking the optical fiber in 1% APES for 1 h, and Figure 8b is the 500 times magnified side view of the APES film generated by dripping with 3.0 mL 1% APES onto the optical fiber surface. Compared to the soaking method, the APES film generated by the dripping method was more uniform and thinner, which was conducive to the detection of other reagents in the later steps.

## 4. Conclusions

In summary, an optical fiber-based MZI for APES detection was proposed. When APES was dripped onto the hydroxyl-modified MZI, the condensation reaction affected the effective refractive index of the cladding mode, which in turn caused a change in the resonant wavelength’s transmission intensity. Measuring the transmission intensities of the MZI revealed that increasing the amount of APES caused the transmission intensities to increase at the beginning and tend toward stabilization with the gradual saturation of the condensation reaction. The optimal amount of 1% APES for fiber-optic biosensors was 3.0 mL, while the optimal amount of 2% APES was 1.5 mL. Given its advantage of easy fabrication, the core-offset-based MZI structure has potential applications for many types of fiber-optic sensors. In addition, in the field of fiber-optic sensing, APES-modified fiber-optic biosensors with amino groups can be widely used in combination with different functional groups for the immobilization of proteins or nucleic acids.

## Figures and Tables

**Figure 1 sensors-21-05870-f001:**
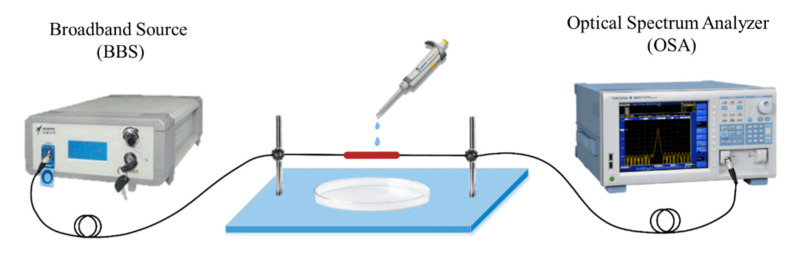
Schematic diagram of the APES fiber-optic sensor.

**Figure 2 sensors-21-05870-f002:**
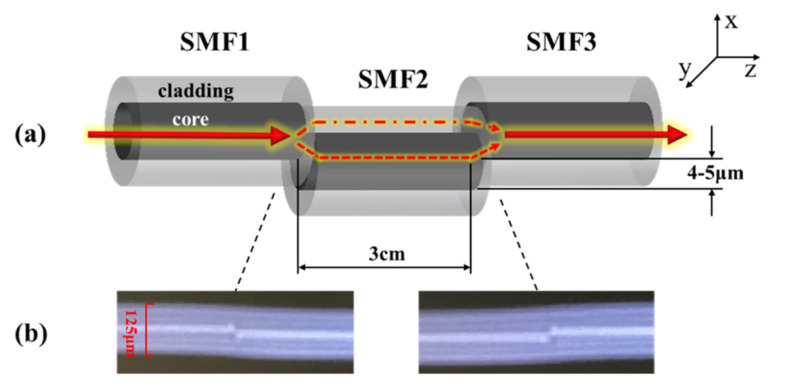
(**a**) The partially enlarged drawing of the MZI (**b**) Pictures of the core-offset connect joints showing on the fusion splicer screen.

**Figure 3 sensors-21-05870-f003:**
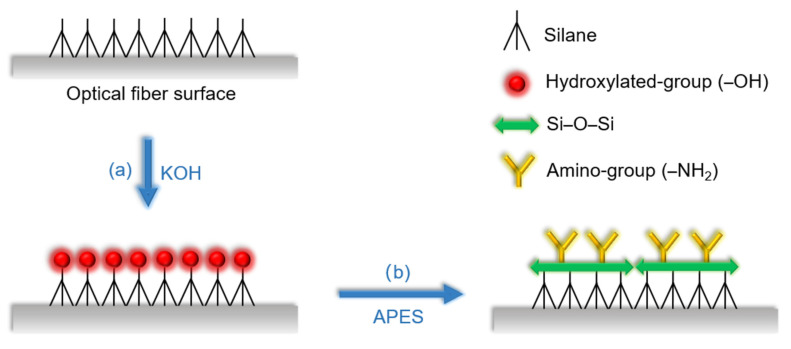
Schematic diagram of (**a**) the hydroxylation process by KOH (**b**) the silanization process by APES.

**Figure 4 sensors-21-05870-f004:**
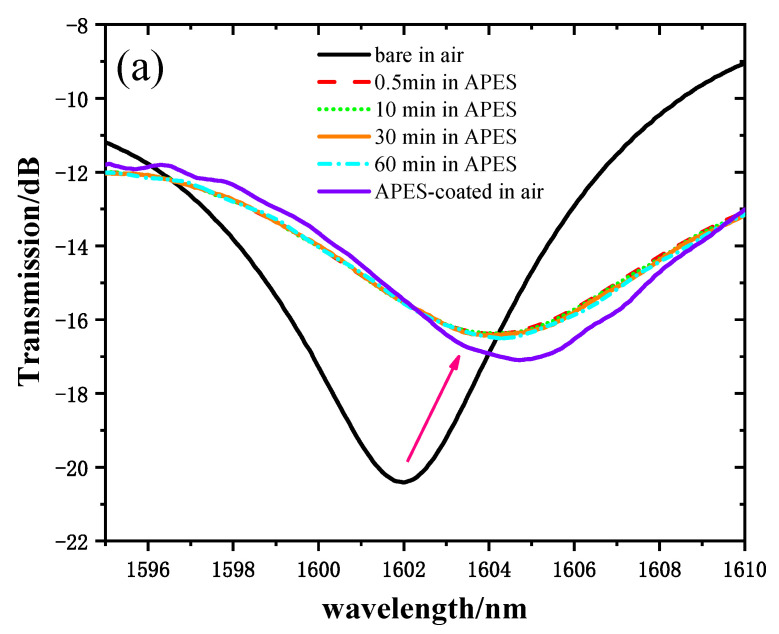
(**a**) Transmission spectra of MZI corresponding to immersion time in 1% APES. (**b**) The relationship between MZI immersion time in 1% APES and transmission intensities.

**Figure 5 sensors-21-05870-f005:**
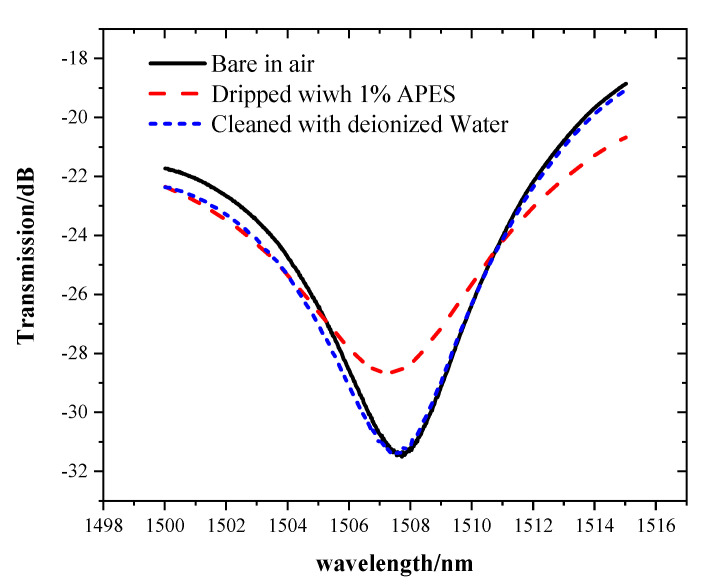
The transmission spectra of the MZI without hydroxyl group.

**Figure 6 sensors-21-05870-f006:**
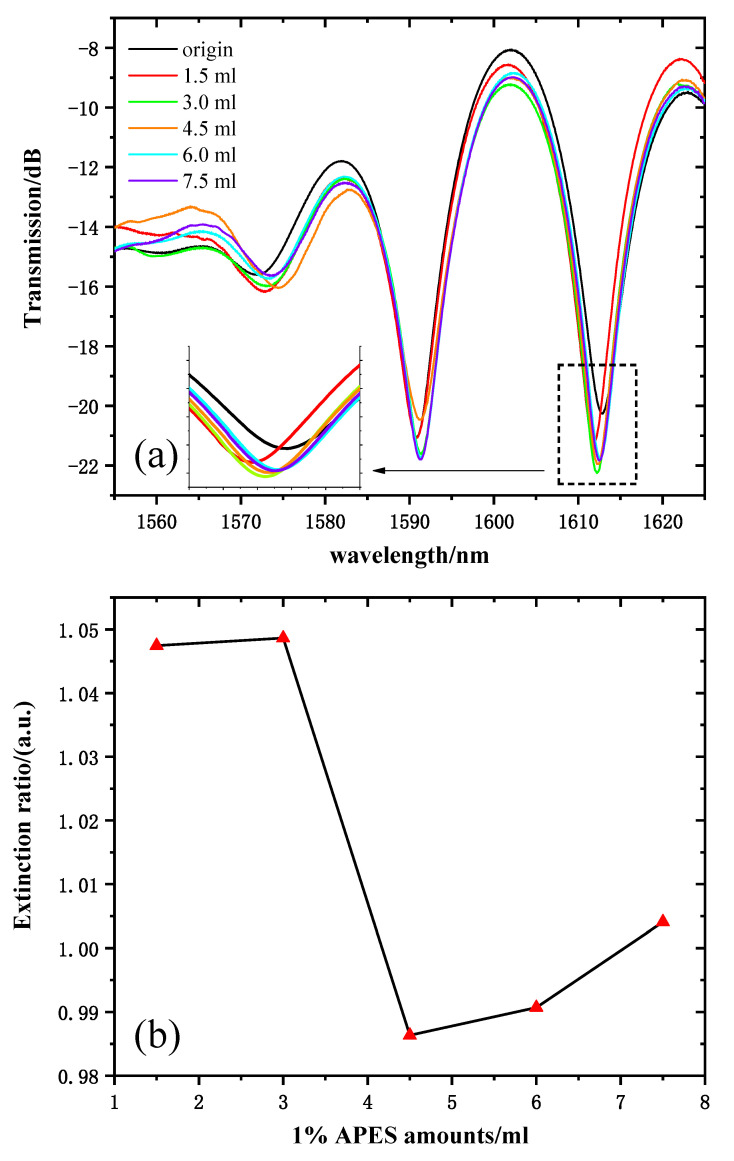
(**a**) MZI transmission spectra corresponding to the MZI dripped with 1% APES. (**b**) The relationship between the amount of 1% APES and the extinction ratio.

**Figure 7 sensors-21-05870-f007:**
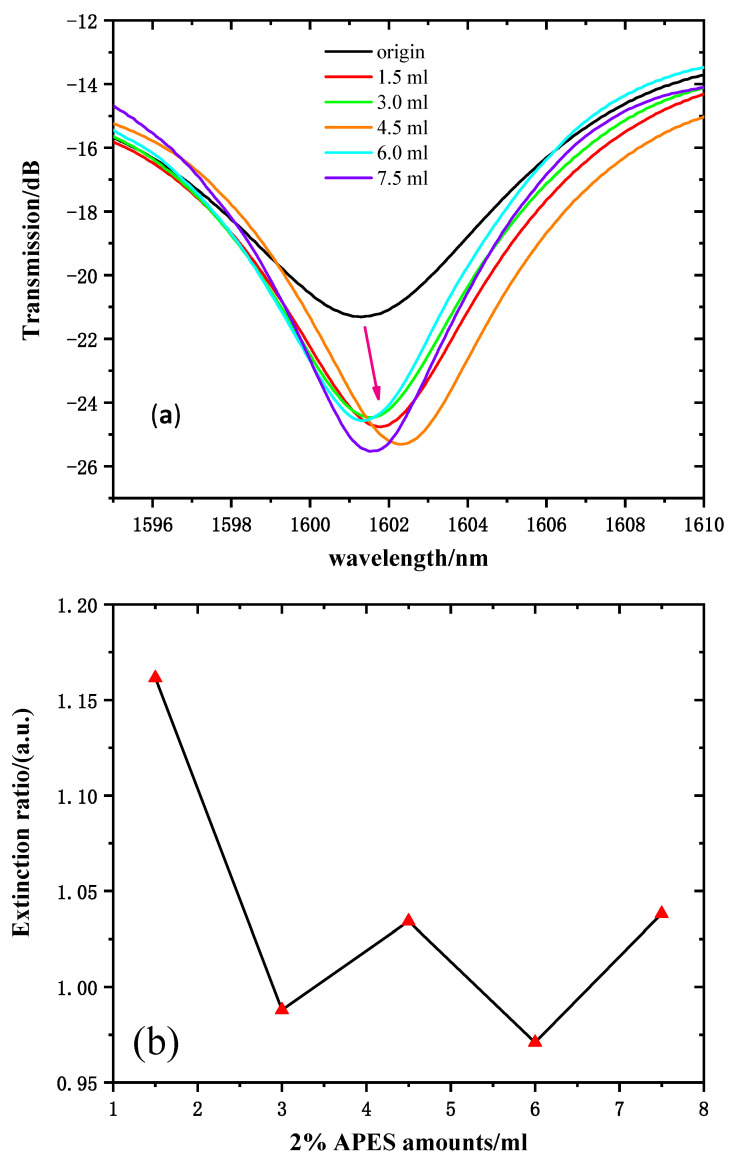
(**a**) Transmission spectra of MZI with fiber dripped with 2% APES. (**b**) The relationship between the amount of 2% APES and the extinction ratio.

**Figure 8 sensors-21-05870-f008:**
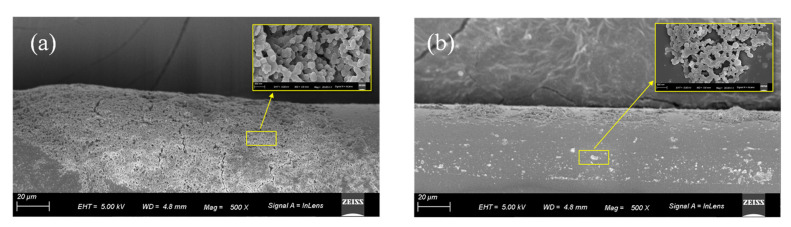
SEM micrographs detailing morphologies of APES film on the optical fiber surface: (**a**) soaked with 1% APES for 1 h; (**b**) dripped with 3 mL 1% APES. Inset: 5k times magnified view of the APES film.

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
