# Peer review of "Optical Fiber Based Mach-Zehnder Interferometer for APES Detection"

_sensors, 2021, doi:10.3390/s21175870_

Round 1

Reviewer 1 Report

In this work, the authors propose an optical fiber sensor for detecting APES based on a MZI.  The author focuses on finding the optimized amount of the APES toluene solution. However, some details are missed.

  1. Authors should provide full names for abbreviations that first appear in the manuscript, such as “SMF” in the abstract section.
  2. There are some grammatical errors format problems (no space between numbers and units) in the manuscript, please check them carefully and change them.
  3. The author should give the ruler scale in Fig. 2b.
  4. There is no explanation of Fig. 5 in the manuscript. Although the process of getting Fig. 5 is described, the phenomenon in Fig. 5 is not explained.
  5. The author did not give a complete period of the interference spectrum, and in Fig. 6b, 7b, the extinction ratio of the transmission spectrum should be used as the ordinate instead of the absolute intensity.
  6. Is the data in Fig. 7a obtained through a device? If it is, as the amount of APTES increases, the light intensity changes, which proves that the chemical bonding is still going on, and the phase shift of the interference spectrum should show a regular change as Sensors & Actuators: B. Chemical 277 (2018) 353–359. Obviously, the data provided by the author here does not show this regularity. Please specify.
  7. In Fig. 7b, the result was not performed on a solution smaller than 1.5ml, but the authors declare the 1.5ml as the optimal concentration. The experimental results obtained are not convincing.
  8. As can be seen from a larger view of Fig. 8b, it is not uniform. In addition, the author should include a clearer ruler scale to facilitate observation.
  9. In the manuscript, the authors present the interferometer and find the best use of APES silanization. So, is this data useful for optical fiber processing of other structures? Or is it the same for different MZI lengths of the interferometer? For example, 1 cm, 2 cm, 4 cm, 5 cm?
  10. What is its repeatability, stability, sensitivity and response time as a sensor?

Reviewer 2 Report

1) Although measurements in fiber interferometers are sensitive to temperature I am completely missing any disputation on this both in the theoretical and experimental part of the paper.

2) The presence of a sample in the measuring arm of the interferometer affects both the resonant wavelength and attenuation (or transmission intensity). It may be beneficial for the readers if the authors also present a graph showing the relationship between the sample amount and the resonant wavelength in a simillar way they are doing it with the intensity in figures 6 and 7.

Round 2

Reviewer 1 Report

The paper was carefully revised and I believe it can be published now.